# Simultaneous Detection and Analysis of Free Amino Acids and Glutathione in Different Shrimp

**DOI:** 10.3390/foods11172599

**Published:** 2022-08-26

**Authors:** Yinzhe Jin, Minhua Xu, Yingshan Jin, Shanggui Deng, Ningping Tao, Weiqiang Qiu

**Affiliations:** 1College of Food Science and Technology, Shanghai Ocean University, Shanghai 201306, China; 2College of Bioscience and Technology, Yangzhou University, Yangzhou 277600, China; 3College of Food and Pharmacy, Zhejiang Ocean University, Zhoushan 316000, China

**Keywords:** shrimp, free amino acids, glutathione, simultaneous detection, automatic amino acid analyzer

## Abstract

An amino acid analyzer method for the simultaneous determination of 20 free amino acids (FAAs) and glutathione (GSH) in *Penaeus vannamei* (PV), *Penaeus vannamei*, *Penaeus hidulis* (PH) and *Penaeus japonicus* (PJ) were developed. The effects of different concentrations of trichloroacetic acid (TCA) and ethanol on the extraction of free amino acids were investigated, and 120 g·L^−1^ TCA was found to be ideal. The target analytes were eluted in sodium citrate buffer B1 (pH = 3.3) containing 135 mL·L^−1^ ethanol and 1 mol·L^−1^ sodium hydroxide (7 mL) and at the optimizing conversion time of sodium citrate buffer B2 (pH = 3.2) and sodium citrate buffer B3 (pH = 4.0) of 5.6 min, and the effective separation was achieved within 29.5 min. The developed method showed good linearity (R^2^ ≥ 0.9991) in the range of 1–250 µg·mL^−1^ with good intra-day and inter-day precision (relative standard deviations ≤ 2.38%) and spike recovery (86.42–103.64%). GSH and cysteine were used to identify marine prawn and freshwater shrimp. Hydroxyproline and serine were used to distinguish PV and *Macrobrachium nipponense* (MN) from others, respectively. The highest content of the total FAAs was found in PV, and principal component analysis revealed that PV had the highest comprehensive score for FAAs and GSH. Arginine was found to have the greatest influence on shrimp flavor.

## 1. Introduction

Free amino acids (FAAs) play an important role in the freshness and flavor of fish products, ornithine can be used as an indicator of freshness of aquatic products [1,2]. Glutamic acid (Glu) and aspartic acid (Asp) are important fresh taste substances, while alanine (Ala) and serine (Ser) contribute to sweetness [1,2]. The flavor of blue mussel was the strongest in summer with more umami-enhancing FAAs [3]. Ala, Ser, and glycine (Gly) were known to be key factors in influencing the sweetness of grass carp [4]. At present, the study of FAAs and methods for their analysis were mostly concentrated on plants [5,6,7] and terrestrial animal-based foods [8,9], while only a few studies focus on aquatic products, especially the different varieties of shrimp. Glutathione (GSH) is a tripeptide composed of cysteine (Cys), Glu, and Gly. It has antioxidation [10], detoxification, and other physiological functions, and it participates in amino acid transport and absorption, in addition to adding flavor to food. The analysis of GSH has mostly been conducted in fruits [11] and plasma [12,13].

Shrimp, one of the fastest-growing aquaculture species [14], refers to a wide variety of arthropod crustaceans. These include marine shrimp, such as *Penaeus vannamei* (PV) and *Penaeus monodon* (PM) [15,16], and freshwater shrimp, such as *Exopalaemon modestus* (EM) and *Macrobrachium nipponense* (MN) [17,18]. The market and nutritional values of shrimp [19] are high because they are rich in protein, FAAs, and minerals; low in fat; and appealing to the taste. Due to different living environments, the contents of FAAs and biologically active peptides in shrimp are also different. Therefore, studying the components and contents of FAAs and GSH in the shrimp from different ecosystems is of great significance.

The currently used analysis methods for FAAs and GSH mainly include capillary electrophoresis [20], high performance liquid chromatography (HPLC) [5,21], liquid chromatography with tandem mass spectrometry (LC-MS/MS) [6,8,22,23], and automatic amino acid analyzer [7,24,25]. Capillary electrophoresis has the limited stability of electroosmotic flow and the poor reproducibility of the dissociated state of the inner silicon capillary wall. Although HPLC is simple to operate, the sample requires pre-column derivatization, which requires the strict control of the derivatization conditions, and thus the stability of the results is easily affected. LC-MS/MS is another detection method, but it requires stringent analytical conditions and is particularly demanding in terms of sample pretreatment and mobile phase. An automatic amino acid analyzer is commonly used to detect amino acids using a cation exchange resin and acid buffers as the stationary and mobile phases, respectively. After column separation, ninhydrin solution is used to yield amino acid derivatives for their detection by visible light absorption. This method has the advantages of simple sample preparation, high automation, good repeatability, and reliable results and less influence by the stability of derivatives [25]. However, the currently used automatic amino acid analyzer can only detect the composition of 18 FAAs [26], and the used methods are mainly limited to the detection of FAAs or GSH. In particular, there is no rapid and effective method for the simultaneous extraction, characterization, and quantification of FAAs and GSH in marine and freshwater shrimp.

In this study, FAAs and GSH contents were simultaneously measured by a chromatographic method, particularly the selection of an extractant, the optimization of the composition of the buffer solution, and elution procedure. For the first time, this method was applied for the measurement of FAAs and GSH profiles in different shrimp varieties. The measured results were analyzed using analysis of variance (ANOVA) and principal component analysis (PCA).

## 2. Materials and Methods

### 2.1. Materials and Instruments

Fished from the sea, live *Penaeus vannamei* (PV) and *Penaeus monodon* (PM) (with an average weight of 15–20 g) were purchased from Luchaogang Market (Shanghai, China) and placed in an oxygenated sampling vessel. Farmed in freshwater, live *Exopalaemon modestus* (EM) and *Macrobrachium nipponense* (MN) (average weight 10–15 g) were obtained from Henanfei Food Store (Shanghai, China). Among them were a total of 95 PV; a total of 5 PM; a total of 5 MN; and a total of 5 EM. Five parallel experiments per shrimp species were carried out.

For sample preparation, all shrimp were quickly transported to the lab within 30 min, and upon arrival, the shrimp were immediately placed on ice, washed with cold water, and blotted dry. The experiment was approved by the Animal Care and Use Committee of Shanghai Ocean University (SHOU-DW-2021-096).

Seventeen types of mixed amino acid standard solution were purchased from Wako Pure Chemical Corporation (Osaka, Japan), taurine (Tau), and GSH from Sigma-Aldrich China (Shanghai, China); L-hydroxyproline (Hyp) from Yuanye Biological Technology (Shanghai, China); and ornithine (Orn) and trichloroacetic acid (TCA) from Maclean Biochemical Technology (Shanghai, China). Analytical grade sodium citrate dihydrate, sodium hydroxide, sodium chloride, citric acid, ethanol, hydrochloric acid (HCl), octoic acid, and polyoxyethylene lauryl ether (Brij-35) were obtained from Sinopharm Chemical Reagent (Shanghai, China).

A DS-1 high-speed tissue masher was purchased from Shanghai Specimen Model Factory (Shanghai, China), a Sartorius BSA124S electronic balance from Sartorius Scientific Instruments (Beijing, China), a H1850R desktop high-speed refrigerated centrifuge from Xiangyi Centrifuge Instrument Co., Ltd. (Shanghai, China), and a T10 basic ULTRA-TURRAX homogenizer from IKA (Staufen, Germany). These instruments were used for sample preparation. A LA8080 amino acid automatic analyzer (Hitachi High-Technologies Corporation, Tokyo, Japan) was used to separate and detect FAAs and GSH in the samples. The water used in the experiment was obtained from a Milli-Q IQ7000 ultra-pure water purification system (Merck Chemical Technology Co., Ltd., Shanghai China). A Mettler-Toledo S2 pH meter (Mettler-Toledo Instruments Co., Ltd., Shanghai China) was used to determine the pH of the sample solution.

### 2.2. Chromatographic Conditions

The parameters of the automatic amino acid analyzer are as follows:

The LA8080 automatic amino acid analyzer (Hitachi, Japan) used ion exchange chromatography separation and post-column derivatization of ninhydrin. The amino acids of the sample were separated in the separation column and then transported by the buffer to the reaction unit for derivatization with the ninhydrin solution. Stationary phase and separation column: cation exchange resin column with a particle size of 3 μm (i.d. 4.6 mm × 60 mm); Separation column temperature 57 °C. Detection wavelength: 570 nm and 440 nm; Mobile phase: sodium citrate buffer B1, B2, B3, B4, and B5 (as shown in Table 1). The total flow rate of sodium citrate buffer B1, B2, B3, B4, and B5 was kept constant at 0.4 mL·min^−1^ (as shown in Table 2); Injection volume: 20 µL.

The reaction unit: ninhydrin reaction solutions, R1, R2, and R3, were prepared with compositions as follows: R1 contained 39 g of ninhydrin, 81 g of sodium borohydride, and 979 mL of propylene glycol monomethyl. R2 consisted of 204 g sodium acetate, 123 g glacial acetic acid, 401 mL propylene glycol monomethyl ether, and 336 mL of ultra-pure water. R3 comprised 50 mL ethanol and 950 mL of ultra-pure water. Ninhydrin reaction solution flow rate was kept constant at 0.35 mL·min^−1^ (as shown in Table 2), reaction unit temperature 135 °C.

The optimization of the elution procedure and heating process are shown in Table 2.

### 2.3. Preparation of Standard Stock Solution

GSH (2.5 µmol·mL**^−^**^1^) standard solution was prepared by dissolving 76.8 mg GSH in 80 mL of 0.2 mol·L**^−^**^1^ HCl, and then the solution was made up to 100 mL using HCl. By dissolving 16.5 mg, 16.4 mg, and 15.6 mg of Orn, Hyp, and Tau, respectively, in 0.2 mol·L**^−^**^1^ HCl, 2.5 µmol·mL**^−^**^1^ standard solutions (50 mL) were prepared. A mixed amino acid stock solution (0.1 µmol·mL**^−^**^1^) was prepared from 2 mL of 2.5 µmol·mL**^−^**^1^ GSH, Orn, Hyp, Tau, and 17 amino acids standard solutions, made up to 50 mL with 0.2 mol·L**^−^**^1^ HCl and stored at −20 °C.

### 2.4. Sample Preparation

A mashed shrimp muscle sample (2.0 g) was weighed into a 50 mL centrifuge tube, and 15 mL of the extractant (12% TCA) was added. After sufficient homogenization (3 min), the sample was centrifuged at 10,610× *g* for 20 min with the temperature maintained at 4 °C. The supernatant was collected and diluted to 50 mL with the same extractant. A 5 mL aliquot was transferred into a centrifuge tube, neutralized with sodium hydroxide (1 mol·L**^−^**^1^) to a pH 2.2 ± 0.02 and diluted to 15 mL with ultra-pure water. The resulting solution was filtered through a 0.22 µm aqueous phase filtration membrane, and the filtrate was injected into an amino acid automatic analyzer for testing. The remaining sample was stored at −20 °C and was frozen in time for the next injection. Extraction experiments were all carried out with five parallel.

### 2.5. Method Validation

Linearity range, the limit of detection (LOD), the limit of quantitation (LOQ), intra-day and inter-day precision, spike recovery, and accuracy were considered in method validation. The linearity range, LOD, and LOQ were detected by diluting the mixed standard stock solution (0.1 µmol·mL**^−^**^1^) with 0.02 mol·L**^−^**^1^ HCl to a concentration range of 1–250 µg·mL**^−^**^1^. The intra-day and inter-day precisions were obtained by adding the mixed standard stock solution to the shrimp sample and detecting five replicates, while the spike recovery was calculated by detecting high, medium, and low concentrations of the mixed standard stock solution (1–50 µmol·mL**^−^**^1^) added to shrimp samples.
Spike recovery (%) = (detected amount − original amount)/spiked amount × 100(1)

### 2.6. Statistical Analysis

ANOVA is a statistical tool used to evaluate the statistical significance of the survey or experimental data for different classes or groups. PCA is a multivariate statistical technique that is most widely used for data exploration, pattern recognition, dimensionality reduction, and data visualization. In this study, ANOVA and PCA were employed using SPSS version 18.0 software (US) to identify the presence of any significant differences (*p* < 0.05) and account for the correlations that exist between the FAAs and GSH data of the four types of shrimp. The different shrimp varieties were comprehensively evaluated using PCA. The FAAs and GSH contents were expressed as mean ± standard deviation.

## 3. Results and Discussion

### 3.1. Optimization of Chromatographic Conditions

FAAs with similar properties are difficult to separate, especially the co-eluting species of Gly, Ala, and Cys. Moreover, only 17 FAAs could be separated and detected using the original buffer composition and elution procedure.

The composition of buffer solution B1 was changed by increasing the ethanol content of B1 from 100 mL·L^−1^ to 160 mL·L^−1^ and sodium hydroxide. When ethanol content was low, Ala and Cys could not be completely separated, while 160 mL·L^−1^ ethanol caused the poor separation of Gly and Ala. As shown in Figure 1a, 20 FAAs and GSH were basically separated in 135 mL·L^−1^ ethanol. Figure 1b,c show that when 7 mL of 1 mol·L^−1^ sodium hydroxide was added to B1, the elution time of the amino acids was reduced by 1 min, and the effect on separation improved. The optimum concentrations of ethanol and sodium hydroxide in B1 improve the elution efficiency, and the stationary phase of the column in this method is cation exchange resin. An appropriate ethanol concentration can effectively clean the column, and an appropriate sodium ion concentration can effectively regenerate the cation exchange resin and improve the separation effect of the column as well as extend the column life.

The elution procedure was also optimized. In particular, the timing for switchover from buffer solutions B2 to B3 gradually increased from 4.5 to 6.5 min. As shown in Figure 1a, when the timing for switchover from B2 to B3 was 5.6 min, 21 target analytes were eluted faster, and the analysis efficiency was higher. The elution procedure following the optimization were shown in Table 2. Compared with the original method, which eluted only 17 amino acids, all 20 FAAs and GSH were eluted within 29.5 min by the optimized method.

### 3.2. Selection of Extractant

The extractants commonly used to extract FAAs, GSH, and precipitated proteins are TCA and ethanol [20,27,28]. Different extractant concentrations also affected the extraction amount. In this experiment, PV, a more common shrimp with a higher nutritional value, was selected as a standard addition experiment to compare the extraction effects of the different concentrations of TCA and ethanol as the extractants. When 120 g·L^−1^ TCA and 500 mL·L^−1^ ethanol were used as extractants, the extraction effect was the greatest (Figure 2). When 500 mL·L^−1^ ethanol was used, there were many impurity peaks and large errors (Figure 2), perhaps due to the side chain of the amino acids which affects the solubility in mixed solvents, such as ethanol and water [29], resulting in the excessive dissolution of some amino acids in ethanol. This may be due to the principle of similar compatibility, as the target analytes were easily dissolved when the polarities of FAAs, GSH, and TCA at the corresponding concentrations were similar. When the concentration of TCA was increased, a greater number of acid soluble substances were dissolved, resulting in a reduction in the extraction amount of the target analytes. Therefore, 120 g·L^−1^ TCA was selected as the extraction solvent.

### 3.3. Method Validation

#### 3.3.1. Mixed Standard Solution and Sample Chromatogram

The basic principle of the L-8800 automatic amino acid analyzer is to wash out FAAs (acidic, neutral, and alkaline) with different acid buffer solutions. FAAs were usually derivatized with ninhydrin, which can produce blue-purple or yellow compounds. The amino group in GSH reacts with ninhydrin in an acidic solution to form a blue-purple compound, which has a maximum absorption peak at 570 nm. The reaction of ninhydrin with hydroxyproline does not produce NH_3_ but directly produces yellow compound with an absorption wavelength at 440 nm. Therefore, FAAs and GSH can be analyzed both qualitatively and quantitatively. As shown in Figure 3, FAAs and GSH in the standard mixed solution and four types of shrimp achieved a better separation effect after optimization.

#### 3.3.2. Linearity and Detection Limit

The mixed stock standard solution of each concentration was determined using three replicates, as described in Section 2.5. The retention time was taken as a qualitative basis. The working curve was prepared based on the peak area of the measured chromatogram and its corresponding mass concentration. Linearity was evaluated using the liner correlation Coefficient of determination (R^2^). Using the signal-to-noise ratio (S/N), the concentration at S/N = 3 is the limit of detection, i.e., the amino acid peak height is approximately three times higher than the baseline noise level. S/N = 10 is the limit of quantification, i.e., when the amino acid peak height is approximately 10 times higher than the baseline noise level First, prepare a lower concentration of the amino acid control solution, inject it into the amino acid analyzer, and observe how many times its peak height is higher than the baseline noise (assume X times), and dilute the solution to X/3 times, which is basically the detection limit of the substance, and dilute the solution to X/10 times, which is basically the quantitative limit of the substance.

As shown in Table 3, all 20 FAAs and GSH had good linear relationships in the range of 1–250 µg·mL^−1^. The R^2^ value was 0.9991 or higher. The LOQ values were between 0.21–2.40 µmol·L^−1^, while the LOD values were between 0.07 and 0.73 µmol·L^−1^. The ranges of linearity and sensitivity were similar to that of previously published size exclusion chromatography with mass spectrometry methods [30]. Therefore, the optimized automatic amino acid analyzer method had good linearity and sensitivity for the determination of FAAs and GSH.

#### 3.3.3. Precision and Recovery

Intra-day and inter-day variations were estimated to evaluate the precision of this method. The relative standard deviation (RSD) was obtained by multiplying the ratio of the standard deviation and the measured amount by 100. The target analytes of intra-day and inter-day RSD were in the range of 0.31–0.71% and 1.23–2.38%, respectively, as shown in Table 3. The spike recovery was between 86.42 and 103.64% for 20 FAAs and GSH. The good precision and recovery indicated that the developed method was suitable for the simultaneous analysis of FAAs and GSH in shrimp samples.

### 3.4. Analysis of FAAs and GSH in Different Shrimp

#### 3.4.1. Analysis of FAAs Profile and GSH Content

To test the applicability of this method, four species of shrimp were analyzed to evaluate the differences in FAAs composition and GSH content, the results of which are given in Table 4. As expected, PV was detected with 20 amino acids, the most diverse among the four shrimp samples, PM and EM were detected with 19 and 18 amino acids, respectively, while only 17 amino acids were found in MN. The results indicated that all four shrimp species were rich in FAAs. The highest content of total free amino acids (TFAAs) and essential amino acids (EAAs) were observed in PV, followed by *Penaeus monodon*, whereas the contents of EM and MN were much lower. EM and MN were not significantly different in TFAAs but the EAAs content of MN was higher, indicating that it may provide more nutrients.

In terms of amino acids abundance, most of FAAs in PV and PM was significantly different (*p* < 0.05), except for Ser, Ala, and Orn. The main FAAs of the four types of shrimp were Gly, proline (Pro), arginine (Arg), and Ala, with a proportion of 85.91% in PV, 89.2% in PM, 90.07% in EM, and 84.89% in MN. These results were in good agreement with those of previous studies [31,32,33], but the levels of TFAAs measured in this study were relatively high. The content of Ser (103.33 ± 0.76 mg/100 g) in MN was higher than in other shrimp varieties. Tau plays an important role in physiological functions, including antioxidant, immune response regulation, calcium transport, myocardial contraction, retinal development, and endocrine functions [34]. Notably, Tau was found in all shrimp cultivars, and its content in marine prawns was much higher than that in freshwater prawns. This may be due to a significant difference in the synthesis of Tau in the different varieties of marine and freshwater shrimp [34]. Cys cannot be detected in freshwater shrimp, which may be due to its chemical lability and conversion to Tau under the action of cysteine sulphinate decarboxylase. Hyp was only found in PV, which may be due to the degradation of collagen [35] or the production of high-content Pro through hydroxylation [36].

GSH has the effect of prolonging the shelf-life of fish and strengthens its flavor, antioxidation, and other physiological effects. However, it was only detected in marine shrimp, with a content of 13.76 ± 0.14 mg/100 g in PV and 9.54 ± 0.27 mg/100 g in PM. This may be due to the adjustment of osmotic pressure through active peptides, such as glutathione, which allowed marine shrimp to survive in seawater environment.

#### 3.4.2. Taste Effect and Taste Activity Value of Amino Acids

Taste activity value (TAV) was calculated as the ratio between the concentrations of the component molecules in the shrimp muscle and its threshold value measured in simple matrix [26,37,38]. The sample matrix was used to calculate the TAV indexes. The amino acid with TAV > 1 was considered as the active compound contributing to the taste of the sample. The larger the TAV, the more significant is the influence on taste. Due to its relatively low threshold, the low content of amino acids in shrimp muscle may still have a certain effect on flavor. Therefore, the TAV is an effective index for comprehensively analyzing the flavor impact of FAAs.

As shown in Table 5, the amino acid composition of TAV > 1 in the four types of shrimp was roughly the same, all of which consisted of Arg, Gly, and Ala. There were six amino acids in PV with TAV > 1, followed by Arg > Gly > Pro > Glu > Ala > His. PM and EM had five, MN had only four in this order: Arg > Gly > Ala > His. This indicated that the taste of shrimp was closely related to the amino acids. Arg, Gly, Pro, Glu, and Ala contributed greatly to the taste of PM and EM, for which the TAV values in PM were 12.57, 8.92, 3.52, 1.84, and 2.4, respectively, and those in EM were 6.31, 6.32, 1.76, 1.2, and 5.46, respectively. The amino acids of TAV > 1 in PV were more than those in others, indicating that it had the most influence on taste compared to other shrimp varieties. Ala (TAV = 5.46) was the most significant taste contributor to EM, in that it was higher than in other shrimp varieties. Ala, with a small side chain, which was specially dedicated to binding with its receptor, generated a taste of sweetness [39]. The TAV values of His (bitter amino acid) in PV and MN were both greater than 1 and far less than other sweet amino acids, such as Arg, producing a suppressed bitterness [40]. Thus, it can be concluded that Arg had the greatest influence on the flavor of shrimp, with its TAV in the range of 6.31–17.61. Generally, sweet amino acids were major contributors to the taste of shrimp; hence, they could be accredited for the good flavor of shrimp.

#### 3.4.3. PCA and Comprehensive Evaluation of Different Species of Shrimp

The PCA results obtained by analyzing the first two principal components are shown in the loading plots in Figure 4 and principal component scores (Table 6). As shown in Figure 4, the two principal components extracted by PCA had a cumulative variance contribution rate of 98.68% (PC1 = 87.32% and PC2 = 11.36%), which basically reflected the initial information of all variables. PC1 was primarily contributed to by GSH (0.997) and 17 amino acids, such as phenylalanine (Phe) (0.988), Leu (0.885), and Orn (−0.762). PC2 was mainly related to His (0.946), Ser (0.717), and Met (−0.481).

Two new comprehensive indices (*F*_1_ and *F*_2_) obtained by PCA were used to analyze amino acids and GSH instead of the original 21 indices. The two principal components linear relationships of FAAs and GSH in the shrimp were as follows:*F*_1_*=* 0.049X_1_ + 0.013X_2_ + 0.005X_3_ +…+ 0.379X_19_ + 0.008X_20_ + 0.741X_21_(2)
*F*_2_*=* 0.037X_1_ + 0.003X_2_ + 0.002X_3_ +…+ 0.608X_19_ + 0.014X_20_ − 0.637X_21_(3)
*F* = 0.873*F*_1_ + 0.117*F*_2_(4)

Based on the PCA results, taking the relative contribution rate of variance corresponding to each principal component as the weight, a comprehensive evaluation model (*F*) was established, and the comprehensive scores of each sample were calculated. These scores reflected the comprehensive quality of FAAs and GSH in the sample. Owing to the higher contents of Ala and Orn, the scores of EM were much lower than those for the other shrimp. The scores of PV were higher than those of others because of a greater abundance of GSH, Phe, Gly, Pro, Tau, and so on. Therefore, Table 6 shows that the comprehensive quality of FAAs and GSH follow the order: PV > PM > MN > EM.

## 4. Conclusions

In this study, a simple and fast amino acid analysis method for the simultaneous extraction and determination of FAAs and GSH in shrimp with 12% TCA was established. This method was shown to have a good linear range, precision, recovery, LOD, and LOQ, and thus can be used to effectively measure 20 FAAs and GSH in shrimp.

All the four species of shrimp investigated were rich in FAAs, and their main FAAs were Gly, Pro, Arg, and Ala. The content of TFAAs and EAAs followed the order: PV > PV > MN > EM. Only a small amount of GSH was detected in marine shrimp. GSH and Cys were used to identify marine shrimp and freshwater shrimp. Hyp could be used to distinguish PV from other shrimp, and Ser could be used to identify MN from other shrimp. Sweet amino acids had a significant impact on the taste of shrimp muscle, particularly Glu, Arg, Gly, Pro, and Ala. Arg was found to have the greatest influence on the flavor of shrimp.

PCA extracted two principal components from 20 types of FAAs and GSH, with a cumulative contribution rate of 98.68%, which can better reflect the comprehensive information on the quality of shrimp FAAs and GSH. A comprehensive evaluation model, *F* = *0.873F_1_* + *0.117F_2_*, was developed using PCA. According to the comprehensive analysis, the comprehensive quality of FAAs and GSH followed the order: PV > PM > MN > EM.

## Figures and Tables

**Figure 1 foods-11-02599-f001:**
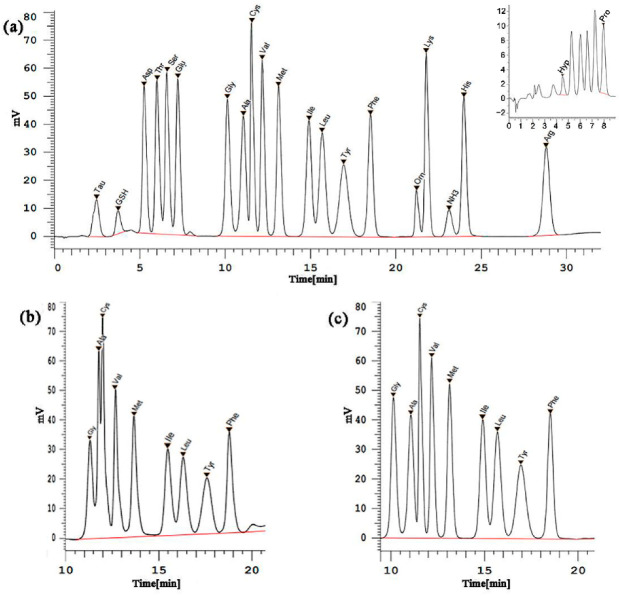
Chromatogram before and after optimization (**a**) Chromatogram of the mixed standard stock solution of 20 FAAs and GSH; (**b**) Chromatogram of buffer B1 on channel 1 without adding sodium hydroxide; (**c**) Chromatograms of buffer B1 on channel 1 adding 7 mL sodium hydroxide (1 mol·L^−1^).

**Figure 2 foods-11-02599-f002:**
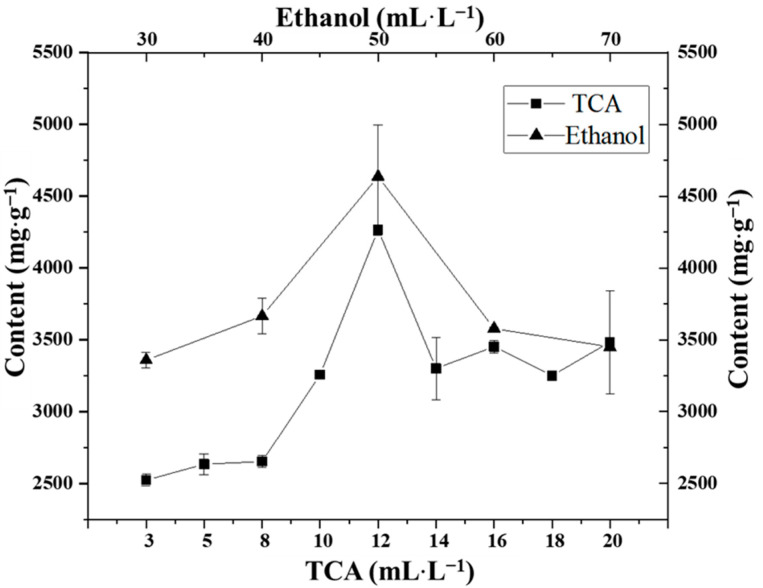
Effects of different extractants on the extraction of FAAs and GSH from PV.

**Figure 3 foods-11-02599-f003:**
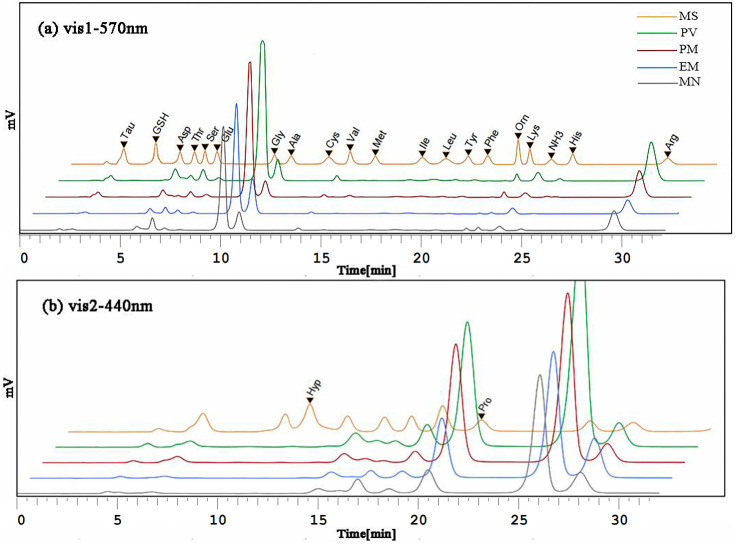
Chromatogram of FAAs and GSH of four types of shrimp. MS: mixed standard stock solution of 20 FAAs and GSH; PV: *Penaeus vannamei*; PM: *Penaeus monodon*; EM: *Exopalaemon modestus*; MN: *Macrobrachium nipponense*. (**a**) The detection wavelength was 570 nm for the other 18 FAAs and GSH; (**b**) The detection wavelength was 440 nm for Hyp and Pro.

**Figure 4 foods-11-02599-f004:**
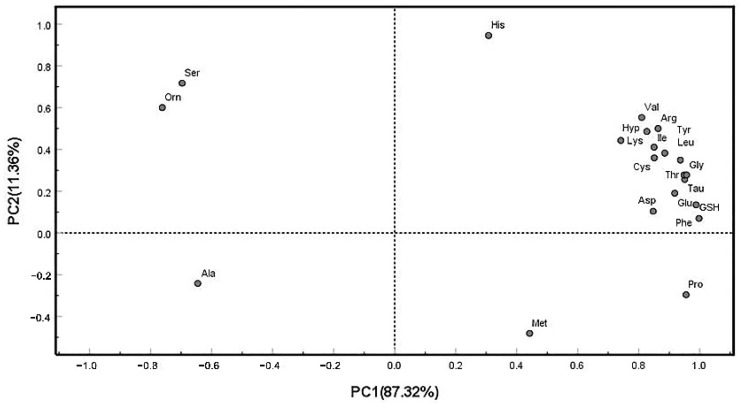
Loading scatter plot for FAAs and GSH contents in principal component analysis (PCA).

**Table 1 foods-11-02599-t001:** Preparation of buffer solution.

Reagent	B1	B2	B3	B4	B5
pH (nominal)	3.3	3.2	4.0	4.9	-
Sodium citrate·2H_2_O (mL)	6.19	7.74	13.31	26.67	-
NaOH (mL)	7.00	15.00	4.00	-	8 g (solid)
NaCl (g)	5.66	7.07	3.74	54.35	-
Sodium citrate·H_2_O (mL)	19.80	22.00	12.80	6.10	-
Ethanol (mL)	135.00	25.00	9.00	-	100.00
Brij-35 (mL)	4.00				
Octoic acid (mL)	0.10				
Ultra-pure water constant Volume (L)	1.00				

**Table 2 foods-11-02599-t002:** Gradient elution and ramp-up procedure after condition optimization of automatic amino acid analyzer.

Time (min)	B1 (%)	B2 (%)	B3 (%)	B4 (%)	B5 (%)	Separation Column Temperature (°C)	R1 (%)	R2 (%)	R3 (%)
0	100	0	0	0	0	57	50	50	0
2.7	100	0	0	0	0		-	-	-
2.8	0	100	0	0	0		-	-	-
5.5	0	100	0	0	0		-	-	-
5.6	0	0	100	0	0		-	-	-
13.6	0	0	100	0	0		-	-	-
13.7	0	0	0	100	0		-	-	-
30.8	0	0	0	100	0		-	-	-
30.9	0	0	0	0	100		-	-	
33.7	0	0	0	0	100		50	50	0
33.8	0	0	0	0	100		0	0	100
34.0	0	0	0	0	100		-	-	-
34.8	0	100	0	0	0		-	-	-
35.9	0	100	0	0	0		-	-	-
38.1	100	0	0	0	0		-	-	-
39.0	100	0	0	0	0		0	0	100
39.1	100	0	0	0	0		50	50	0
55.0	100	0	0	0	0		-	-	-

**Table 3 foods-11-02599-t003:** Correlation coefficient of determination, linear range, the limit of quantitation (LOQ), the limit of detection (LOD), spike recovery, and precision of the mixed stock standard solution of 20 free amino acids (FAAs) and glutathione (GSH).

Analytes	Correlation Coefficient of Determination (R^2^)	Linear Range (µmol·L^−1^)	LOQ (µmol·L^−1^)	LOD (µmol·L^−1^)	Average Spike Recovery (%)	Precision (%RSD, *n* = 5)	
						Intra-Day	Inter-Day
Tau	0.9992	0.40~58.20	0.38	0.10	90.42	0.46	1.23
GSH	0.9994	0.35~30.21	0.30	0.12	88.26	0.51	1.82
Asp	0.9993	0.65~60.32	0.60	0.20	94.25	0.52	1.63
Thr	0.9997	0.75~64.72	0.52	0.22	95.38	0.53	1.68
Ser	0.9991	0.74~65.85	0.74	0.24	87.61	0.71	2.10
Glu	0.9999	0.86~68.92	0.83	0.23	92.33	0.71	2.35
Gly	0.9993	0.85~67.62	0.85	0.27	100.80	0.38	1.32
Ala	0.9997	0.91~78.64	0.91	0.25	87.31	0.36	1.26
Cys	0.9992	1.12~75.82	0.67	0.16	87.92	0.67	2.38
Val	0.9993	1.05~73.16	0.94	0.25	86.42	0.46	1.53
Met	0.9996	0.65~68.59	0.54	0.12	89.57	0.53	1.85
Ile	0.9994	0.85~81.57	0.76	0.23	99.25	0.62	2.10
Leu	0.9992	0.72~83.24	0.60	0.13	102.33	0.31	1.63
Tyr	0.9996	0.54~63.84	0.21	0.07	95.79	0.56	1.94
Phe	0.9993	0.61~68.43	0.43	0.16	103.64	0.61	2.32
Orn	0.9991	0.52~56.16	0.52	0.19	95.10	0.37	1.74
Lys	0.9993	0.47~55.63	0.47	0.12	89.32	0.46	1.83
His	0.9998	0.84~68.37	0.80	0.27	87.67	0.35	1.82
Arg	0.9997	0.82~88.49	0.65	0.19	87.56	0.39	1.42
Hyp	0.9994	0.80~79.65	0.72	0.18	99.51	0.52	1.82
Pro	0.9992	2.40~98.36	2.40	0.73	92.19	0.62	2.01

**Table 4 foods-11-02599-t004:** FAAs profile and GSH of different shrimp.

Component		PV		PM		EM		MN	
		Content (mg/100 g)	Proportion/%	Content (mg/100 g)	Proportion/%	Content (mg/100 g)	Proportion/%	Content (mg/100 g)	Proportion/%
1	Tau	67.63 ± 0.35 ^a^	1.60	34.36 ± 2.39 ^b^	1.02	8.77 ± 0.88 ^c^	0.39	11.22 ± 0.13 ^c^	0.50
2	GSH	13.76 ± 0.14 ^a^	-	9.54 ± 0.27 ^b^	-	ND	-	ND	-
3	Asp	7.37 ± 0.07 ^a^	0.17	2.42 ± 0.13 ^b^	0.07	2.18 ± 0.03 ^c^	0.09	ND	-
4	Thr ^▲^	158.92 ± 1.58 ^a^	3.76	103.2 ± 6.78 ^b^	3.08	57.52 ± 0.88 ^d^	2.60	65.19 ± 0.18 ^c^	2.92
5	Ser	38.41 ± 0.38 ^c^	0.90	11.59 ± 0.65 ^d^	0.34	45.49 ± 0.85 ^b^	2.05	103.33 ± 0.76 ^a^	4.66
6	Glu	106.59 ± 0.95 ^a^	2.52	55.01 ± 3.66 ^b^	1.64	35.79 ± 0.56 ^c^	1.62	25.34 ± 0.16 ^d^	1.14
7	Gly	1479.21 ± 13.37 ^a^	35.01	1158.97 ± 72.05 ^b^	34.64	821.39 ± 14.24 ^d^	37.18	900.71 ± 6.83 ^c^	40.62
8	Ala	195.12 ± 1.58 ^c^	4.61	143.99 ± 9.03 ^d^	4.30	327.47 ± 6.1 ^a^	14.82	216.19 ± 1.2 ^b^	9.73
9	Cys	27.44 ± 2.79 ^a^	0.64	5.08 ± 0.27 ^a^	0.15	ND	-	ND	-
10	Val ^▲^	36.64 ± 0.38 ^a^	0.86	20.43 ± 1.34 ^b^	0.61	13.08 ± 0.36 ^c^	0.59	19.67 ± 0.29 ^b^	0.89
11	Met ^▲^	7.07 ± 0.13 ^b^	0.16	21.31 ± 1.51 ^a^	0.63	5.55 ± 0.34 ^b^	0.25	6.39 ± 0.22 ^b^	0.29
12	Ile ^▲^	15.41 ± 0.29 ^a^	0.36	9.38 ± 0.64 ^b^	0.28	7.58 ± 0.19 ^c^	0.34	8.22 ± 0.18 ^c^	0.37
13	Leu ^▲^	25.85 ± 0.22 ^a^	0.61	16.4 ± 1.09 ^b^	0.49	12.22 ± 0.44 ^d^	0.55	13.54 ± 0.23 ^c^	0.61
14	Tyr	23.51 ± 0.25 ^a^	0.55	15.54 ± 0.71 ^b^	0.46	3.99 ± 0.22 ^c^	0.18	8.96 ± 0.22 ^d^	0.41
15	Phe ^▲^	18.3 ± 0.36 ^a^	0.43	12.89 ± 0.83 ^b^	0.38	5.62 ± 0.24 ^c^	0.25	5.55 ± 0.22 ^c^	0.25
16	Orn	4.87 ± 0.03 ^c^	0.11	4.79 ± 0.18 ^c^	0.14	6.99 ± 0.09 ^b^	0.31	14.8 ± 0.05 ^a^	0.66
17	Lys ^▲^	48.26 ± 0.53 ^a^	1.14	38.12 ± 2.62 ^b^	1.13	9.98 ± 0.11 ^d^	0.45	30.02 ± 0.18 ^c^	1.35
18	His	22.92 ± 0.17 ^a^	0.54	10.5 ± 0.69 ^c^	0.31	3.98 ± 0.03 ^d^	0.18	21.73 ± 0.02 ^b^	0.97
19	Arg	880.04 ± 8.46 ^a^	20.83	628.18 ± 42.91 ^b^	18.77	315.03 ± 4.93 ^d^	14.26	533.58 ± 4.63 ^c^	24.09
20	Hyp	11.51 ± 1.61 ^a^	0.27	ND	-	ND	-	ND	-
21	Pro	1076.86 ± 10.47 ^a^	25.49	1053.77 ± 66.8 ^a^	31.49	526.14 ± 9.6 ^b^	23.81	231.4 ± 2.22 ^c^	10.45
EAAs		310.49 ± 3.28 ^a^	7.34	221.75 ± 14.84 ^b^	6.62	111.58 ± 2.59 ^d^	5.05	148.61 ± 1.53 ^c^	6.72
TFAAs		4250.95 ± 36.96 ^a^	100	3346.00 ± 214.36 ^b^	100	2208.88 ± 38.41 ^c^	100	2215.95 ± 17.8 ^c^	100

Note: Tau, taurine; Asp, aspartic acid; Thr, threonine; Ser, serine; Glu, glutamic acid; Gly, glycine; Ala, alanine; Cys, cysteine; Val, Valine; Met, methionine; Ile, isoleucine; Leu, leucine; Tyr, tyrosine; Phe, phenylalanine; Orn, ornithine; Lys, lysine; His, histidine; Arg, arginase; Hyp, hydroxyproline; Pro, proline; ^▲^ denotes EAAs, essential amino acids; TFAAs, total free amino acids; ND, not detected. Mean ± standard deviation, *n* = 3. Values within rows with non-overlapping superscript letters are significantly different at *p* < 0.05.

**Table 5 foods-11-02599-t005:** Taste attributes (+pleasant, -unpleasant), taste threshold, and taste activity values (TAVs) of the FAAs of different shrimp.

FAAs	Taste Attribute	Taste Threshold (mg/100 g)	TAV			
			PV	PM	EM	MN
Asp	Umami (+)	100	0.08	0.03	0.03	ND
Glu	Umami (+)	30	3.56	1.84	1.20	0.85
Thr	Sweet (+)	260	0.62	0.40	0.23	0.26
Ser	Sweet (+)	150	0.26	0.08	0.31	0.69
Gly	Sweet (+)	130	11.38	8.92	6.32	6.93
Ala	Sweet (+)	60	3.26	2.40	5.46	3.61
Arg	Bitter/sweet (+)	50	17.61	12.57	6.31	10.68
Pro	Sweet/bitter (+)	300	3.59	3.52	1.76	0.78
Val	Sweet/bitter (−)	40	0.92	0.52	0.33	0.50
Met	Bitter/sweet/sulfurous (−)	30	0.24	0.72	0.19	0.22
Ile	Bitter (−)	90	0.18	0.11	0.09	0.10
Leu	Bitter (−)	190	0.14	0.09	0.07	0.08
Phe	Bitter (−)	90	0.21	0.15	0.07	0.07
Lys	Sweet/bitter (−)	50	0.97	0.77	0.20	0.61
His	Bitter (−)	20	1.15	0.53	0.20	1.09

Note: The sample matrix was used to calculate the TAV indexes.

**Table 6 foods-11-02599-t006:** Principal component scores and comprehensive assessment in different shrimp.

Class	F_1_	F_2_	F	Order
PV	2.29	1.20	2.14	1
PM	0.96	−0.40	0.80	2
EM	−1.67	−1.24	−1.60	4
MN	−1.59	0.44	−1.34	3

## Data Availability

Data are contained within the article.

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
