# Peer review of "Simultaneous Detection and Analysis of Free Amino Acids and Glutathione in Different Shrimp"

_foods, 2022, doi:10.3390/foods11172599_

Round 1
Reviewer 1 Report
I would like to appreciate authors to conduct study entitled "Simultaneous Detection and Analysis of Free Amino Acids and Glutathione in Different Shrimp". This study will be positive addition in the field the first time, as a new method was developed for the measurement of FAAs and GSH profiles in different shrimp varieties. I have certain concerns regarding this manuscript that need to be addressed before publication.
1. Line 12-13, scientific name should be written in Italics.
2. Line 21, mention full form of PV and MN.
3. Line 65, rewrite sentence for better understanding.
4. Table 1, 2, 3, 4, mention units of various parameters in brackets.
5. In various tables, wherever applicable, explain abbreviations used in tables in footnote of the respective table.
6. Line 154, spell out the name of membrane.
7. Line 171, mention which version of SPSS you have used along with its country of origin.
8. Figure 2, it seems like you wish to write "Content (mg/100g)" in the left legend instead "Contant".
Author Response
Dear Editors and Reviewers:
Thank you for the reviewers’ useful comments, which give us a big help for the later research. The manuscript has been revised accordingly, modified parts have been marked in red, and the detailed corrections are listed below point by point:
Reviewers' comments:
Q: Line 12-13, scientific name should be written in Italics.
Answer: Article has been revised (line12-13).
Q: Line 21, mention full form of PV and MN.
Answer: Article has been revised (line 12, 22).
Q: Line 65, rewrite sentence for better understanding.
Answer: The modified sentence is that This method has the advantages of simple sample preparation, high automation, good repeatability, and reliable results and less influence by the stability of derivatives (line 64-65).
Q: Table 1, 2, 3, 4, mention units of various parameters in brackets.
Answer: Article has been revised. (Table 1, 2, 3, 4).
Q: In various tables, wherever applicable, explain abbreviations used in tables in footnote of the respective table.
Answer: The article has been supplemented with footnotes after certain tables (line 328-332, 362).
Q: Line 154, spell out the name of membrane.
Answer: The membranes in the article are aqueous phase filter membranes (line 144).
Q: Line 171, mention which version of SPSS you have used along with its country of origin.
Answer: The SPSS version used is SPSS 18.0 (US) (line 162).
Q: Figure 2, it seems like you wish to write "Content (mg/100g)" in the left legend instead "Contant".
Answer: In Figure 2, “Contant” has been modified to “Content”. (line 235).
The revised manuscript has been resubmitted to the journal. We are looking forward to the positive response.
Yours sincerely,
Ningping Tao and Weiqiang Qiu

Reviewer 2 Report
The paper “Simultaneous Detection and Analysis of Free Amino Acids and Glutathione in Different Shrimp” show an interesting method to quantify amino acids in shrimp samples, whit an optimization procedure to improve the chromatographic separation also of Gly, Ala, and Cys. I think that the work deserves to be published, but there are several points that must be clarified.
First of all, the authors assert several times thorough the text their method is “automated”. However, the described method is a chromatographic one, which can be only partially automated, and it also requires a sample preparation procedure. I suggest either to better explain in which way the method can be defined “automated” or to remove such term and replace it simply with “chromatographic”.
The most important point, however, is that the experimental procedure is not clearly described, there are several details missed which must be added.
Following, my remarks about the paper.
In general, when you show a concentration, it is better to write it as, for example mL L-1, rather than mL/L, without the slash. Moreover, it would be better to express all of them ovel liters, rather than over 100mL as they are often shown in the text.
Please, avoid as much as possible abbreviations in the abstract, or at least explain all of them. Abbreviations B2 and B3, for example, are not clear at all. The abstract should be read also without the main text, so it has to be clear.
Lines 38-40: rephrase as: “It has antioxidation [10], detoxification, and other physiological functions, it participates in amino acid transport and absorption, in addition to adding flavor to food.”
Line 65: The sentence “reliable results affected by the stability of derivatives less” is not clear, explain it better or rephrase it
Line 71-72: rephrase this sentence, the verb is missing
Lines 78-86: How may shrimps/samples did you purchased for each species? How many replicates of each sample/species did you analyzed? Add these details in the text.
Lines 107-110: the chromatographic conditions are not clear at all. Which column and stationary phase did you use? Which mobile phase? Which instrument and detector? Please, explain it better
Lines 113-115: Why did you used such unusual volumes? How did you take “979 mL”, “401 mL”, and “336 mL” with such precision?
Line 116: flow rate of what? Did you use a mixture of the buffer and ninhydrin solutions as mobile phase for the HPLC? It is not clear at all, explain it better.
Lines 140-142: again, why did you use such unusual weights with such precision?
Line 149: which extractant?
Line 177: correct “co-elution” with “co-eluting species”.
Lines 187-188: how can you assert that this method can prolong the column lifetime? Add a reference or explain it better.
Figure 1 is not clear at all. It is not clear what you want to show (in particular with figures b) and c)), the FAAs names over the peaks are too small, and the general quality of the image should be improved
Figure 2: correct “Contant” with “Content” or, better, “Concentration” in the axes label. The same for Table 4.
Line 223: “yellow violet color”? What does it mean?
Line 240-242: explain better hot did you calculate LOD and LOQ.
Lines 244-245: correct “LOD (LOQ) value was…” with “LOD (LOQ) values were…”.
Lines 247-248: correct “shrimp species of shrimps” with “shrimp species”.
Line 269: correct “were PV” with “were observed in PV”.
Line 273: correct “difference” with “different”.
Table 4: the number are reported with too many significant digits. In general, it is better to report means and standard deviations with 2 or 3 significant digits (e.g. 1479.21 becomes 1480, 13.76 becomes 13.8). Moreover, so many significant digits would indicate that the authors quantified FAAs at the level of hundredths of a milligram, which is at least unlikely.
Lines 317-318: explain better “water or a simple matrix”. Which sample did you use to calculate the TAV indexes? If water, I think you should have sampled the water in which shrimps were fished or farmed. Was that the case?
Figure 4: please, increase the quality of the figure and the character of the FAAs names.
Lines 353-363: this procedure is not clear at all. First of all, what are the X you used to calculate the factors. Then, how did you used the variances? Moreover, why did you used the PCA results to calculate such indexes? You should know that in a PCA you can add or remove a sample and the results can change dramatically, so using scores and loadings to draw conclusions can be very dangerous. At least show also the scores plot to demonstrate that your PCA is “stable”.
Author Response
Dear Editors and Reviewers:
Thank you for the reviewers’ useful comments, which give us a big help for the later research. The manuscript has been revised accordingly, modified parts have been marked in red, and the detailed corrections are listed below point by point:
Reviewers' comments:
Q: First of all, the authors assert several times thorough the text their method is “automated”. However, the described method is a chromatographic one, which can be only partially automated, and it also requires a sample preparation procedure. I suggest either to better explain in which way the method can be defined “automated” or to remove such term and replace it simply with “chromatographic”.
Answer: We have made correction according to the Reviewer’s comments. We have removed the “automated” and replaced it with “chromatography”. (line 70-71), The name “automatic amino acid analyzer” is the common name of the device manufacturer, so it is not modified.
Q: In general, when you show a concentration, it is better to write it as, for example mL L-1, rather than mL/L, without the slash. Moreover, it would be better to express all of them ovel liters, rather than over 100mL as they are often shown in the text.
Answer: We have re-written it in the full text based on the commenters' suggestions.
Q: Please, avoid as much as possible abbreviations in the abstract, or at least explain all of them. Abbreviations B2 and B3, for example, are not clear at all. The abstract should be read also without the main text, so it has to be clear.
Answer: The abstract abbreviations section of this paper has been supplemented with explanations (line 17, 22).
Q: Line 38-40: rephrase as: “It has antioxidation [10], detoxification, and other physiological functions, it participates in amino acid transport and absorption, in addition to adding flavor to food.”
Answer: Lines 38-40: the statements of the statements of “It has antioxidation [10], detoxification, participates in amino acid transport and absorption, and other physiological functions in addition to adding flavor to food.” were corrected as “It has antioxidation [10], detoxification, and other physiological functions, it participates in amino acid transport and absorption, in addition to adding flavor to food .”(line 38-40).
Q: Line 65: The sentence “reliable results affected by the stability of derivatives less” is not clear, explain it better or rephrase it
Answer: The sentence “reliable results affected by the stability of derivatives less” were corrected as “reliable results and less influence by the stability of derivatives”. (line 64-65).
Q: Line 71-72: rephrase this sentence, the verb is missing
Answer: The sentence “There is no rapid and effective method for the simultaneous extraction, characterization and quantification of FAAs and GSH in marine and freshwater shrimp.” were corrected as “There is no rapid and effective method for the simultaneous extraction, characterization and quantification of FAAs and GSH in marine and freshwater shrimp.” (line 67-69).
Q: Lines 78-86: How may shrimps/samples did you purchased for each species? How many replicates of each sample/species did you analyzed? Add these details in the text.
Answer: Among them, a total of 95 Penaeus vannamei (PV), Penaeus monodon (PM) a total of 5 Macrobrachium nipponense (MN) a total of 5, showed a total of 5 Exopalaemon modestus (EM). Five parallel experiments per shrimp species (line 82-84).
Q: Lines 107-110: the chromatographic conditions are not clear at all. Which column and stationary phase did you use? Which mobile phase? Which instrument and detector? Please, explain it better
Answer: Automatic amino acid analyzer ( LA8080, Hitachi, Japan) uses ion exchange chromatography separation and post-column derivatization of ninhydrin, The amino acids of the sample were separated in the separation column and then transported by the buffer to the reaction unit for derivatization with the ninhydrin solution. Stationary phase and separation column: Cation exchange resin column with a particle size of 3 μm (i.d. 4.6 mm x 60 mm); separation column temperature 57 ℃, Detection wavelength: 570 nm and 440 nm; buffer solution flow rate 0.40 mL·min-1, Mobile phase: sodium citrate buffer B1, B2, B3, B4 and B5 (as shown in Table 1), The total flow rate of sodium citrate buffer B1, B2, B3, B4 and B5 was kept constant at 0.4 mL·min-1 (as shown in Table 2); Injection volume: 20 µL (line 110-118).
The reaction unit: Ninhydrin reaction solutions, R1, R2, and R3, were prepared with compositions as follows: R1 contained 39 g of ninhydrin, 81 g of sodium borohydride, and 979 mL of propylene glycol monomethyl. R2 consisted of 204 g sodium acetate, 123 g glacial acetic acid, 401 mL propylene glycol monomethyl ether, and 336 mL of ultra-pure water. R3 comprised 50 mL ethanol and 950 mL of ultra-pure water. Ninhydrin reaction solution flow rate was kept constant at 0.35 mL·min-1 (as shown in Table 2), reaction unit temperature 135 ℃ (line120-126).
Q: Lines 113-115: Why did you used such unusual volumes? How did you take “979 mL”, “401 mL”, and “336 mL” with such precision?
Answer: The precise dosage of these reagents is recommended by the manufacturer of the automatic amino acid analyzer equipment.
Q: Line 116: flow rate of what? Did you use a mixture of the buffer and ninhydrin solutions as mobile phase for the HPLC? It is not clear at all, explain it better.
Answer: Automatic amino acid analyzer uses ion exchange chromatography separation and post-column derivatization of ninhydrin, The amino acids of the sample were separated in the separation column and then transported by the buffer to the reaction unit for derivatization with the ninhydrin solution. Mobile phase: sodium citrate buffer B1, B2, B3, B4 and B5 (as shown in Table 1) , The total flow rate of sodium citrate buffer B1, B2, B3, B4 and B5 was kept constant at 0.4 mL/min (as shown in Table 2); The reaction unit: Ninhydrin reaction solutions, R1, R2, and R3, Ninhydrin reaction solution flow rate was kept constant at 0.35 mL/min (as shown in Table 2) (line 110-126)
Q: Lines 140-142: again, why did you use such unusual weights with such precision?
Answer: The precise dosage of these reagents is recommended by the manufacturer of the automatic amino acid analyzer equipment
Q: Line 149: which extractant?
Answer: The extractant is 12% TCA (line 139).
Q: Line 177: correct“co-elution”with “co-eluting species”.
Answer: The statement of “co-elution” was corrected as “co-eluting species”. (line 168-169).
Q: Lines 187-188: how can you assert that this method can prolong the column lifetime? Add a reference or explain it better.
Answer: The stationary phase of the column in this method is cation exchange resin. An appropriate ethanol concentration can effectively clean the column, and an appropriate sodium ion concentration can effectively regenerate the cation exchange resin and improve the separation effect of the column as well as extend the column life (line 178-182).
Q: Figure 1 is not clear at all. It is not clear what you want to show (in particular with figures b) and c)), the FAAs names over the peaks are too small, and the general quality of the image should be improved
Answer: The clarity of the figure has been modified (line 214).
Q: Figure2: correct “Contant” with “Content” or, better, “Concentration” in the axes label. The same for Table 4.
Answer: The article has been changed from contant to content, and Table 4 has been changed (line 235, 327).
Q: Line 223: “yellow violet color”? What does it mean?
Answer: Line 223: The statements of “FAAs typically react with ninhydrin to give rise to purple or yellow violet color.” were corrected as “FAAs were usually derivatized with ninhydrin, which can produce blue-purple or yellow compounds.” (line 240-241).
Q: Line 240-242: explain better hot did you calculate LOD and LOQ.
Answer: Calculation method of LOD and LOQ: Using the signal-to-noise ratio (S/N), the concentration at S/N=3 is the limit of detection, i.e., the amino acid peak height is approximately 3 times higher than the baseline noise level. S/N=10 is the limit of quantification, i.e. when the amino acid peak height is approximately 10 times higher than the baseline noise level First, prepare a lower concentration of the amino acid control solution, inject it into the amino acid analyzer, and observe how many times its peak height is higher than the baseline noise (assume X times), and dilute the solution to X/3 times, which is basically the detection limit of the substance, and dilute the solution to X/10 times, which is basically the quantitative limit of the substance (line 258-266).
Q: Lines 244-245: correct “LOD (LOQ) value was…” with “LOD (LOQ) values were…”.
Answer: The statements of “LOD (LOQ) value was…” were corrected as “LOD (LOQ) values were…”. (line 268-269).
Q: Lines 247-248: correct “shrimp species of shrimps” with “shrimp species”.
Answer: The statement of “shrimp species of shrimps” was corrected as “shrimp species”. (line 294).
Q: Line 269: correct “were PV” with “were observed in PV”.
Answer: The statement of “were PV” was corrected as “were observed in PV”. (line 294).
Q: Line 273: correct “difference” with “different”.
Answer: The statement of “difference” was corrected as “different”. (line 299).
Q: Table 4: the number are reported with too many significant digits. In general, it is better to report means and standard deviations with 2 or 3 significant digits (e.g. 1479.21 becomes 1480, 13.76 becomes 13.8). Moreover, so many significant digits would indicate that the authors quantified FAAs at the level of hundredths of a milligram, which is at least unlikely.
Answer: Table 4 has been modified to 2 or 3 significant digits (line 327).
Q: Lines 317-318: explain better “water or a simple matrix”. Which sample did you use to calculate the TAV indexes? If water, I think you should have sampled the water in which shrimps were fished or farmed. Was that the case?
Answer: The statement of “water or a simple matrix” was corrected as “simple matrix”. The sample matrix was used to calculate the TAV indexes (line 336-337).
Q: Figure 4: please, increase the quality of the figure and the character of the FAAs names.
Answer: The clarity of the figure has been revised and the full names of the FAAs abbreviations in the figure are shown in Table 4, so the full names of the FAAs in Figure 4 are omitted (line 371).
Q: Lines 353-363: this procedure is not clear at all. First of all, what are the X you used to calculate the factors. Then, how did you used the variances? Moreover, why did you used the PCA results to calculate such indexes? You should know that in a PCA you can add or remove a sample and the results can change dramatically, so using scores and loadings to draw conclusions can be very dangerous. At least show also the scores plot to demonstrate that your PCA is “stable”.
Answer: The questions raised by the reviewer are indeed very valid, X is for FAA and GSH; adding or subtracting a sample in the PCA analysis in this paper may result in a huge change in the results. The main purpose of applying the PCA method in this paper is to distinguish or identify different species of shrimp by the composition pattern of free amino acids, and we have previously considered the possibility that sample changes may have a great impact on the PCA results. However, we found through this experiment that the cumulative variance contribution of increasing or decreasing a sample principal component was more than 95%, so we expressed it according to this model, and if you think it is not appropriate, we can delete it according to your opinion, thank you!
The revised manuscript has been resubmitted to the journal. We are looking forward to the positive response.
Yours sincerely,
Ningping Tao and Weiqiang Qiu

Reviewer 3 Report
The manuscript deals with the development and validation of a method based on amino acids analyzer for the separation and quantitation of free amino acids, including glutathione, in shrimps. With respect to the conventional conditions, the gradient used for the separation of the analytes was modified to allow for the complete resolution of the amino acids. The method was then applied to real samples (different varieties of shrimps) and the results in terms of amino acid content were treated by PCA to have comprehensive information on the product/food quality.
-On line 21 of the Abstract: the meaning of the abbreviations PV and MN should be given.
-lines 28-40. The sentence should be improved
-line 51. “analysis” instead of “detection”
-line 55. Please take care, the poor separation reproducibility in capillary electrophoresis does not depend on the short optical path, rather is related to the limited stability of electroosmotic flow and the poor reproducibility of the dissociation status of the inner silica capillary wall.
-lines 56-58. Please take care, the sentence should be removed or improved since several inaccuracies are reported. As an example: what is "derivatization speed"? In addition, it is also assumed that the analysis by HPLC should be done by derivatization; however, this is not always the case, thus it would be necessary to properly modify the sentence and then introduce it in the right context.
-lines 58-60. The criticism toward LC-MS/MS should be avoided because it is not really motivated.
-line 140 (and following). It is not correct reporting the weight of a standard using 4 significant figures. It is highly unlikely that the reported quantities can be exactly weighed.
-line 150. The centrifugation speed should be given in "g".
-line 152 and following. It seems that only one extraction per sample is done. Is it right?
-lines 206 and 208. Which solvent?
-lines 209-211. The sentence contains several inaccuracies such as “shape of amino acids” and “activity of chain groups”. It could be “...due to the side chain of the amino acids which affects the solubility.....”
-Fig. 2 (and Table 4). “Content”
-Fig. 3. The x-axis is missing.
-line 240 (and related table). Coefficient of determination
-Table 3. Ser, Gly, Ala, Orn, and Lys. The LOQ values cannot be higher than the lowest calibration point
-line 258. It has been reported that three spiking levels were done, thus it is expected to report the recovery obtained at each of the levels.
-lines 267-268. Improve English.
-Table 7. Please check the data for Gly content. The reported values seem to be out of calibration range; at this very high level, the linearity could not be observed unless a higher range is considered.
Author Response
Dear Editors and Reviewers:
Thank you for the reviewers’ useful comments, which give us a big help for the later research. The manuscript has been revised accordingly, modified parts have been marked in red, and the detailed corrections are listed below point by point:
Reviewers' comments:
Q: On line 21 of the Abstract: the meaning of the abbreviations PV and MN should be given.
Answer: Penaeus vannamei (PV); Macrobrachium nipponense (MN) (line 12, 22).
Q: lines 28-40. The sentence should be improved
Answer: We have made correction according to the Reviewer’s comments (line31-32, 39-41).
Q: line 51. “analysis” instead of “detection”
Answer: Article has been revised (line 51).
Q: line 55. Please take care, the poor separation reproducibility in capillary electrophoresis does not depend on the short optical path, rather is related to the limited stability of electroosmotic flow and the poor reproducibility of the dissociation status of the inner silica capillary wall.
Answer: We have made correction according to the Reviewer’s comments: Capillary electrophoresis has limited stability of electroosmotic flow and poor reproducibility of the dissociated state of the inner silicon capillary wall. (line 54-56).
Q: lines 56-58. Please take care, the sentence should be removed or improved since several inaccuracies are reported. As an example: what is "derivatization speed"? In addition, it is also assumed that the analysis by HPLC should be done by derivatization; however, this is not always the case, thus it would be necessary to properly modify the sentence and then introduce it in the right context.
Answer: We have made correction according to the Reviewer’s comments: Although HPLC is simple to operate, the sample requires pre-column derivatization, which requires strict control of the derivatization conditions, and thus the stability of the results is easily affected. (line 56-58).
Q: lines 58-60. The criticism toward LC-MS/MS should be avoided because it is not really motivated.
Answer: We have made correction according to the Reviewer’s comments: LC-MS/MS is another detection method, but it requires stringent analytical conditions and is particularly demanding in terms of sample pretreatment and mobile phase. (line 58-60).
Q: line 140 (and following). It is not correct reporting the weight of a standard using 4 significant figures. It is highly unlikely that the reported quantities can be exactly weighed.
Answer: Article has been revised (line 130-136).
Q: line 150. The centrifugation speed should be given in "g".
Answer: Article has been revised (line 140).
Q: line 152 and following. It seems that only one extraction per sample is done. Is it right?
Answer: Extraction experiments are all with five parallel (line146-147).
Q: lines 206 and 208. Which solvent?
Answer: The extraction effects of different extractants were compared first, and 12% TCA was determined as the optimal extractant, so the extractant was 12% TCA.
Q: lines 209-211. The sentence contains several inaccuracies such as “shape of amino acids” and “activity of chain groups”. It could be “...due to the side chain of the amino acids which affects the solubility.....”
Answer: We have made correction according to the Reviewer’s comments (line 227-228).
Q: Fig. 2 (and Table 4). “Content”
Answer: Article has been revised (line 235,327).
Q: Fig. 3. The x-axis is missing.
Answer: Article has been revised (line 248).
Q: line 240 (and related table). Coefficient of determination
Answer: Article has been revised (line 258, 274).
Q: Table 3. Ser, Gly, Ala, Orn, and Lys. The LOQ values cannot be higher than the lowest calibration point
Answer: Table 3. We have made correction according to the Reviewer’s comments.
Q: line 258. It has been reported that three spiking levels were done, thus it is expected to report the recovery obtained at each of the levels.
Answer: The spiked recoveries in Table 3 are actually the average spiked recoveries, which have been corrected in the text.
Q: lines 267-268. Improve English.
Answer: Article has been revised (line 290-293).
Q: Table 7. Please check the data for Gly content. The reported values seem to be out of calibration range; at this very high level, the linearity could not be observed unless a higher range is considered.
Answer: The Gly content in shrimp is indeed very high and is beyond the calibration range, so the Gly content will be measured separately and the sample will be diluted 5-10 times more before testing on the machine, and ensure Gly content is within the calibration range
The revised manuscript has been resubmitted to the journal. We are looking forward to the positive response.
Yours sincerely,
Ningping Tao and Weiqiang Qiu
